# Dimensional reduction by geometrical frustration in a cubic antiferromagnet composed of tetrahedral clusters

Ryutaro Okuma [1,2,7 ✉], Maiko Kofu [3], Shinichiro Asai[1], Maxim Avdeev [4,5], Akihiro Koda[6], Hirotaka Okabe[6], Masatoshi Hiraishi[6], Soshi Takeshita [6], Kenji M. Kojima[6,8], Ryosuke Kadono[6], Takatsugu Masuda [1], Kenji Nakajima [3] & Zenji Hiroi [1]

Dimensionality is a critical factor in determining the properties of solids and is an apparent built-in character of the crystal structure. However, it can be an emergent and tunable property in geometrically frustrated spin systems. Here, we study the spin dynamics of the tetrahedral cluster antiferromagnet, pharmacosiderite, via muon spin resonance and neutron scattering. We find that the spin correlation exhibits a two-dimensional characteristic despite the isotropic connectivity of tetrahedral clusters made of spin 5/2 $Fe^{3+}$ ions in the three-dimensional cubic crystal, which we ascribe to two-dimensionalisation by geometrical frustration based on spin wave calculations. Moreover, we suggest that even one-dimensionalisation occurs in the decoupled layers, generating low-energy and one-dimensional excitation modes, causing large spin fluctuation in the classical spin system. Pharmacosiderite facilitates studying the emergence of low-dimensionality and manipulating anisotropic responses arising from the dimensionality using an external magnetic field.

[1] Institute for Solid State Physics, University of Tokyo, Kashiwa, Chiba, Japan. [2] Okinawa Institute of Science and Technology Graduate University, Onna, Okinawa, Japan. [3] Materials and Life Science Division, J-PARC Center, Japan Atomic Energy Agency, Tokai, Ibaraki, Japan. [4] Australian Nuclear Science and Technology Organisation, New Illawarra Road, Lucas Heights, Australia. [5] School of Chemistry, The University of Sydney, Sydney, Australia. [6] Institute of Materials Structure Science, High Energy Accelerator Research Organization (KEK-IMSS), Tsukuba, Ibaraki, Japan. [7] Present address: Clarendon Laboratory, University of Oxford, Parks Road, Oxford, UK. [8] Present address: Center for Molecular and Materials Science, TRIUMF, Vancouver, BC, Canada. ✉ email: ryutaro.okuma@gmail.com

All crystalline solids occur in three-dimensional (3D) space but can attain certain quasi-low dimensionalities arising from anisotropic chemical bonds in their crystal structures. For instance, carbon atoms in diamonds possess a 3D network through $sp^3$ bonding, whereas in graphite, they have a two-dimensional (2D) structure through $sp^2$ bonding. Itinerant or localised electrons in a crystal are affected by such anisotropic arrangements of atoms and exhibit a variety of phenomena depending on dimensionality; diamond is hard and insulating, whereas graphite is soft and semimetallic.

Dimensionality critically governs phase transitions and elementary excitations in electronic crystals. In the classical phase transition scenario, a certain symmetry present at high temperatures is spontaneously broken at low temperatures for 3D systems. In contrast, for lower dimensions, according to the Mermin–Wagner theorem, continuous symmetry cannot be spontaneously broken so that the corresponding long-range ordering is prohibited for non-zero temperatures[1]. Hence, low-dimensional systems can realise states of matter that are different from conventional ones and excitations that go beyond the concept of symmetry breaking. Such a system is approximately materialised in an actual 3D crystal via anisotropic chemical bonds: for example, 2D square lattices made of $Cu^{2+}$ ions in cuprate superconductors[2]. Intriguing properties arising from low-dimensional magnetism have been one of the central issues in condensed matter physics.

Low dimensionality in spin systems can be effectively enhanced by geometrical frustration. Let us consider an anisotropic triangular lattice (ATL) consisting of parallel one-dimensional (1D) spin chains with antiferromagnetic (AF) interactions, $J$, which are connected to their neighbours by inter-chain interactions, $J'$, in a zigzag manner (Fig. 1a). In an ATL antiferromagnet, one-dimensionality becomes predominant even with a sizable $J'$ when AF correlations have developed in the chains at low temperatures because the $J'$ couplings are geometrically cancelled out, resulting in a set of decoupled spin chains. This one-dimensionalisation by frustration is theoretically expected even for large $J'$ values up to $J'/J = \sim0.65$[3–5] and has been experimentally observed in $Cs_2CuCl_4$ and $Ca_3ReO_5Cl_2$ comprising ATLs made of spin-1/2 $Cu^{2+}$ and $Re^{6+}$ ions, respectively, with $J'/J$

$= 0.3–0.4$[6,7]. Moreover, two-dimensionalisation by frustration has been observed for compounds having body-centred tetragonal (BCT) lattices (Fig. 1b) such as $BaCuSi_2O_6$[8,9] and $CePd_2Si_2$[10]. Notably, for these ATL and BCT magnets, the corresponding low dimensionalities are already embedded in the original crystal structures, which create geometrical frustration.

In contrast, magnetic low dimensionality can emerge purely from a highly frustrated 3D network made of regular tetrahedra (a pyrochlore lattice, Fig. 1c)[11]. In such a system, the global energy minimum is achieved by a null total spin on a tetrahedron, and thus, macroscopically-degenerate spin arrangements. When a small-scale interaction or quantum fluctuation picks up a 3D pattern out of the ensemble, a low-dimensional excitation appears as a result of local spin precession modes. This is observed in the chromium spinel $ZnCr_2O_4$ as a zero-dimensional mode confined in the hexagonal ring of the pyrochlore lattice[12]. 1D and 2D correlations were also observed in spinel $ZnV_2O_4$[13] and double perovskite $Sr_2YRuO_6$[14], respectively. Although certain deformations in the lattice drive the systems to magnetic order and cause low-dimensional excitations in some materials, low dimensionality can, in principle, appear without spatial anisotropy of the underlying lattice.

In this paper, we report a unique 3D tetrahedral-cluster antiferromagnet, pharmacosiderite $((H_3O)Fe_4(AsO_4)_3$ $(OH)_4 \cdot 5.5H_2O)$[15,16], in which low dimensionality is not present in the crystal structure but emerges purely by geometrical frustration without lattice distortions.

## Results and discussion

### 3D cluster magnet pharmacosiderite.
Pharmacosiderite with the general formula $AFe_4(AsO_4)_3(OH)_4 \cdot nH_2O$ (A is a monovalent cation) crystallises into a cubic structure comprising clusters made of four $FeO_6$ octahedra (Fig. 2a, Supplementary Note 1).

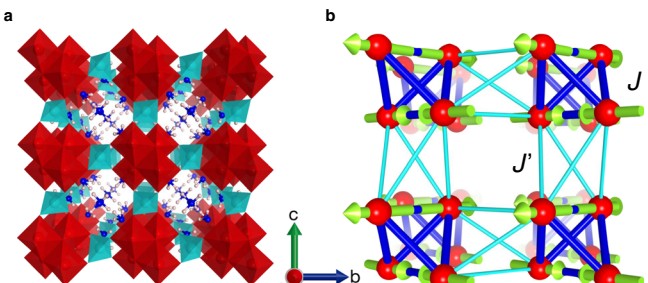

**Fig. 2 Crystal and magnetic structures of pharmacosiderite, $(D_3O)$ $Fe_4(AsO_4)_3(OD)_4 \cdot 5.5D_2O$. a** Cubic crystal structure of the space group $P\text{–}43m$. The red octahedra, sky-blue tetrahedra, blue spheres, and pink spheres represent $FeO_6$ octahedral units, $AsO_4$ tetrahedral units, oxygens, and hydrogens of water molecules, respectively. The red, blue, and green arrows indicate the crystallographic $a$, $b$, and $c$ axes, respectively. Four $FeO_6$ octahedra with small trigonal distortions are connected by their edges to form a cluster of $T_d$ symmetry, forming a regular tetrahedron made of $Fe^{3+}$ ions. These clusters are located at the vertices of the cubic unit cell with ~8 Å on each edge and connected to neighbours via a $[AsO_4]^{3-}$ tetrahedron along the edge of the cube. The cube possesses a large open space at the centre, which accommodates various large monovalent ions, a hydronium ion in this case, and water molecules. **b** Fe sublattice and the $\mathbf{q} = 0$, $\Gamma_5$ magnetic structure below $T_N = 6$ K. Red spheres represent spin-5/2 $Fe^{3+}$ ions, four of which form a regular tetrahedral cluster at each vertex of the cubic unit cell. The yellow green arrows represent the directions of the magnetic moments of Fe spins, all of which lie approximately in the planes perpendicular to the $c$ axis. The thick blue and thin sky-blue sticks represent intracluster Heisenberg interactions $J$ and intercluster Heisenberg interactions $J'$, respectively.

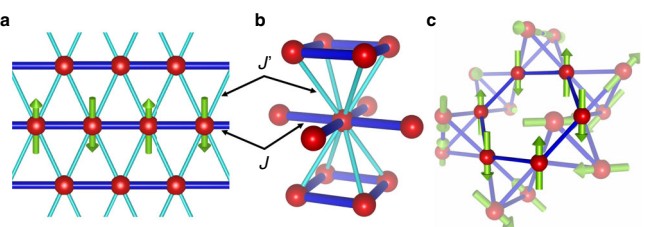

**Fig. 1 Geometrically frustrated lattices.** The red spheres and green arrows represent magnetic ions and spins, and the thick blue and thin sky-blue sticks represent the strong and weak magnetic couplings between them, respectively. **a** Anisotropic triangular lattice composed of 1D chains with spins coupled by AF interaction $J$, which are connected in a staggered manner by $J'$ ($J > J'$). The arrows represent antiferromagnetically aligned spins in the centre chain. Two $J'$ interactions in an isosceles triangle cancel each other, effectively eliminating the inter-chain interactions. **b** Body-centred tetragonal lattice made of 2D square lattices with $J$ stacked in a staggered manner by $J'$. The four $J'$ couplings are geometrically cancelled when AF correlations develop in the plane. **c** 3D pyrochlore lattice made of tetrahedra connected by their vertices. The spin arrangement in $ZnCr_2O_4$ contains a cooperative precession mode of six collinear spins on a hexagonal ring, which behaves as a zero-dimensional excitation because interactions with the surrounding antiparallel spins lead to geometrical cancellation[12].

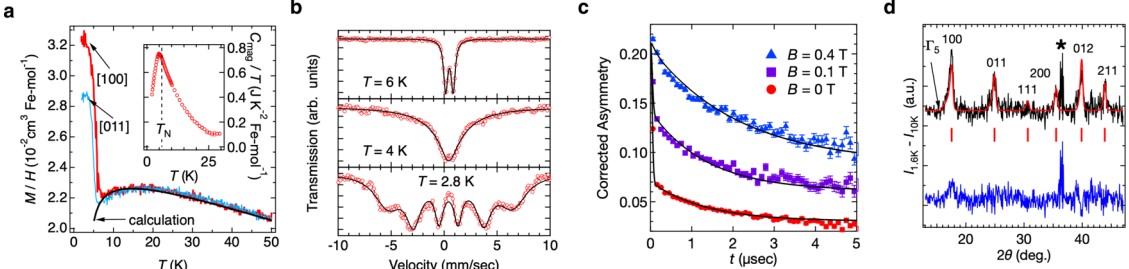

**Fig. 3 Experimental data demonstrating an LRO in pharmacosiderite. a** Temperature dependencies of the magnetisation $M$ divided by magnetic-field $H$ measured under magnetic fields of $\mu_0 H = 0.1$ T applied along the [100] and [011] directions of a single crystal of pharmacosiderite (main panel) and magnetic heat capacity $C_{mag}/T$ at zero field (Inset). The curve on the $M/H$ data represents a fit to a coupled-cluster model, which yields $J = 10.6$ K and $J' = 2.9$ K[16]. **b** Mössbauer spectra of the powder sample of pharmacosiderite at $T = 2.8$, 4, and 6 K[16]. The black line is a fit to a Blume–Tjon model[35] taking into account fluctuating local magnetic fields (Supplementary Note 2). Even below $T_N$, the spectra are broad, and the intensity ratio deviates from the conventional powder average, suggesting magnetic fluctuations remaining in the LRO. **c** Time-dependent $\mu^+$-$e^+$ decay asymmetry, which is proportional to the muon spin polarisation, measured at 2 K in a zero external field and longitudinal fields of 0.1 and 0.4 T, shown by red circle, purple square, and blue triangle, respectively. The error bars indicate the standard deviations of the muon decay asymmetry at each point. The curve on each dataset represents a fit to a model in which one muon site near a hydrogen feels a static local field and the other muon site feels both static and fluctuating local fields (Supplementary Note 3). **d** Powder neutron diffraction pattern and a Rietveld fitting to the $\mathbf{q} = 0$, $\Gamma_5$ spin structure depicted in Fig. 2b. The black line represents data taken at 1.6 K after the subtraction of the 10 K data as a reference of nuclear contributions. The red line, blue line, and green bars represent a Rietveld fit, a residual of fitting, and the positions of magnetic Bragg peaks, respectively. The asterisk * indicates the residual from the subtraction of the 10 K data from the 1.6 K.

The magnetic sublattice consists of regular tetrahedral clusters made of spin-5/2 $Fe^{3+}$ ions arranged in a primitive cubic cell (Fig. 2b); a related lattice with tetrahedra arranged in a face-centred cubic lattice is the breathing pyrochlore lattice[17]. The intra- and intercluster magnetic interactions, $J$ and $J'$, are estimated to be AF with magnitudes of 10.6 and 2.9 K, respectively, from the fitting of the magnetic susceptibility (Fig. 3a)[16]. Notably, there is a strong frustration in both the tetrahedral cluster and the intercluster couplings because the four $J'$ paths form an elongated tetrahedron along the cube edges.

A magnetic long-range order (LRO) sets in below $T_N = 6$ K with a weak ferromagnetic moment of $7 \times 10^{-3}$ $\mu_B$ predominantly along the [100] axis (Fig. 3a). Interestingly, however, previous Mössbauer spectroscopy measurements depicted an broad spectrum even in the LRO state (Fig. 3b)[16]. Such broadening could arise from structural inhomogeneity (Supplementary Note 1). However, it is more likely that the broad spectrum is predominantly caused by the phenomenon in which the ordered spin is constantly flipped at a frequency of ~$10^2$ MHz (Supplementary Note 2). Indeed, our muon spin relaxation (μSR) experiments confirmed such a dynamical fluctuation with a similar time scale in the ordered state, as revealed by Mössbauer spectroscopy (Fig. 3c and Supplementary Note 3). The implanted muon experiences a fluctuating local field overlaid on a static internal field, which is evidenced by the observation that the muon spin exhibits an exponential relaxation of spin polarisation by the fluctuating internal field after an initial recovery due to the static field even under longitudinal fields up to 0.4 T. These results strongly suggest the coexistence of two types of magnetism with different dynamics at the microscopic level. Hence, the LRO may be unconventional, and an intriguing phenomenon is awaiting discovery.

To obtain a deeper insight into the magnetism of pharmacosiderite, we carried out neutron scattering experiments on a polycrystalline sample of $(D_3O)Fe_4(AsO_4)_3(OD)_4\bullet 5.5D_2O$. The magnetic properties of pharmacosiderite are insensitive to the choice of the A-site cation (Supplementary Note 3). Figure 3d shows the magnetic contribution at 1.6 K; a nuclear contribution has been subtracted using the 10 K data as a reference. No structural transition was observed down to 1.6 K (Supplementary

Note 1); hence, the cubic symmetry was preserved, which is in agreement with previous X-ray diffraction experiments[16]. The magnetic peaks have nearly the same widths as the nuclear peaks (Supplementary Note 4), indicative of a long-range magnetic order; the magnetic correlation length estimated is 360 Å, which is 45 times larger than the $a$-axis length.

All the magnetic Bragg peaks appeared in the same positions as the nuclear peaks, indicating a $\mathbf{q} = 0$ magnetic order. Thus, spin arrangements in a single tetrahedron are sufficient to be considered. Because the minimum requirement is to make the total spin zero in each tetrahedron, there are five possible spin arrangements (Supplementary Note 4). Our Rietveld fitting to the data pinned down a $\mathbf{q} = 0$, $\Gamma_5$ magnetic structure, as depicted in Fig. 2b.

The spin arrangement of the $\mathbf{q} = 0$, $\Gamma_5$ magnetic order is coplanar in the (001) plane with two pairs of antiparallel spins in one tetrahedron aligned along the two edges perpendicular to the $c$ axis; it is a type of two-in, two-out structure with local [110] anisotropy and not [111] anisotropy as in the spin-ice compounds[18]. A small canting of each spin is possible along the $c$ axis, resulting in a minuscule net moment when a weak single-ion anisotropy exists along the local [111] direction. Notably, the magnetic structure is essentially tetragonal, but we observed no tetragonal lattice distortion, probably because of negligible spin–lattice couplings owing to the indirect structural connection between clusters via the $AsO_4$ units, which is different from the case of spinel compounds[11].

A notable finding of the magnetic structure analysis is that the magnitude of the refined magnetic moment at 1.6 K is 2.08(4) $\mu_B$, which is much smaller than the expected value of 5 $\mu_B$ for a spin-5/2 system; the ordered moment is almost saturated at this temperature, judging from the temperature evolution of the (100) peak intensity (Supplementary Fig. 10c). This significant reduction in the ordered magnetic moment is consistent with the persistent fluctuations observed in the μSR and Mössbauer spectroscopy. However, we should note that the large nuclear $R_{wp}$ factor of ~27%, which mostly comes from partial deuteration, suggests that the standard deviation of the ordered moment value derived from the Rietveld analysis is somewhat underestimated (Supplementary Note 1).

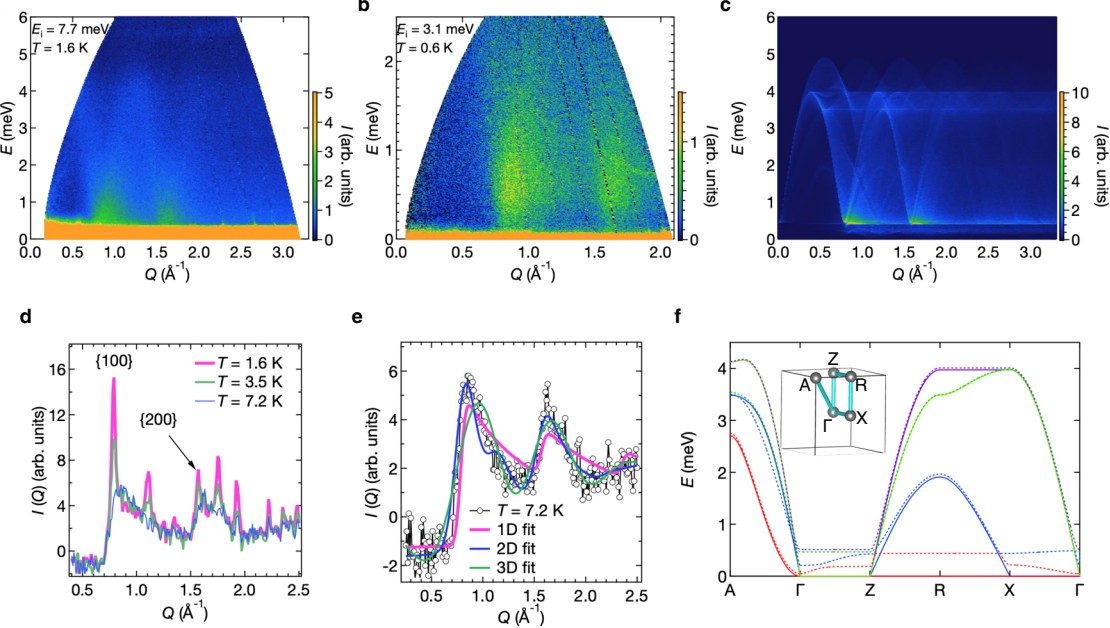

**Fig. 4 Inelastic neutron scattering experiments from a powder sample of pharmacosiderite and simulations based on the linear SW theory. a, b** Powder-averaged equal-time structure factors $S(Q, E)$ measured at $T = 1.6$ K and an incident neutron energy of $E_i = 7.7$ meV (**a**) and at $T = 0.6$ K and $E_i = 3.1$ meV (**b**). The colour scales of the neutron intensity are shown on the right side of (**c**). **c** Simulated powder-averaged $S(Q, E)$ calculated by the linear SW theory for the **q** = 0, $\Gamma_5$ magnetic structure. The isotropic interactions $J = 0.9$ meV and $J' = 0.27$ meV, which were determined from magnetic susceptibility measurements[16], are assumed. In addition, an intracluster DM interaction of $0.01 J$ and an easy-axis single-ion anisotropy of $0.001 J$ along [111] are included in the calculation. **d** Temperature evolution of the integrated structure factor $I(q)$ deduced from the $S(Q, E)$ after the subtraction of the data collected at 50 K as a reference for the lattice contribution (see text). The magenta, green, and blue lines represent the data obtained at 1.6, 3.5, and 7.2 K, respectively. **e** Fitting of the $I(q)$ at 7.2 K to the low-dimensional scattering model. The magenta, green, and blue lines represent fits for 1D, 2D, and 3D scattering, respectively. The agreement factors $R_{wp}$ are 6.0%, 3.3% and 6.6%, respectively, indicating that the 2D model best describes the data. See Supplementary Note 7 for details of the fits. **f** Dispersion relation of magnetic excitations calculated for a single crystal. SWs between the selected high symmetry $k$ points in the Brillouin zone shown in the inset are calculated by the linear SW theory for the **q** = 0, $\Gamma_5$ magnetic structure with the same parameters used for (**c**). The solid and dash-dotted lines represent the dispersions without and with anisotropic interactions, respectively. As there are four sites in the magnetic unit cell, four modes distinguished by different colours appear.

**Inelastic neutron scattering experiments**. To obtain information on the magnetic excitations and the origin of the spin fluctuations, we performed inelastic neutron scattering experiments. A powder-averaged dynamical structure factor $S(Q, E)$ obtained at an incident neutron energy of $E_i = 7.7$ meV at $T = 1.6$ K is shown in Fig. 4a. Large intensities are observed at approximately $Q/\text{Å}^{-1} = 0.8$ and 1.6, which correspond to the (100) and (200) reflections in the elastic channel, respectively. Dispersive modes develop from these and form domes with maxima at ~4 meV, which are apparently spin-wave (SW) excitations from the **q** = 0 magnetic order. In contrast, an $S(Q, E)$ obtained at an $E_i$ of 3.1 meV and $T = 0.6$ K, which is more sensitive to low-energy excitations, evidences a small gap with a magnitude of 0.5 meV (Fig. 4b and Supplementary Note 5).

The observed $S(Q, E)$s were compared with simulations based on the linear SW theory for the **q** = 0, $\Gamma_5$ structure after convolution with the instrumental resolution (Fig. 4c). The overall features, the bandwidth and intensity distribution, are well reproduced by the calculation with no adjustable parameters; $J = 0.9$ meV and $J' = 0.27$ meV from the fitting of magnetic susceptibility data[16]. Small intracluster interactions were added to account for the gap in the spectrum; a Dzyaloshinskii–Moriya (DM) interaction of $0.01 J$, which is likely to exist and contributes to selecting the **q** = 0 and $\Gamma_5$ order among others, as well as an easy-axis single-ion anisotropy of $0.001 J$, which causes a small spin canting, are necessary to explain the weak ferromagnetism observed by magnetometry. However, the observed spectra are much broader than the calculated ones, especially along the

energy axis (Supplementary Note 6), which is consistent with the reduced ordered moment in the ground state.

To investigate the spin fluctuations in pharmacosiderite, we examined the $S(Q, E)$ data in terms of the equal-time structure factor of spin correlation perpendicular to a scattering vector $\langle S_i^\perp(0) \cdot S_j^\perp(0) \rangle$, which corresponds to a Fourier transform of a spin configurations snapshot. Because most of the intensity is concentrated between 0 and 2 meV, as shown in Supplementary Figs. 11 and 12, the total scattering is approximated by integrating $S(Q, E)$ within this energy range as follows:

$$I(Q) \propto \int_0^{2\text{meV}} S(Q, E)dE \sim \int_{-\infty}^{\infty} S(Q, E)dE = \sum_{ij} e^{iQ(r_i - r_j)}\langle S_i^\perp \cdot S_j^\perp \rangle \tag{1}$$

The temperature evolution of $I(Q)$ is plotted in Fig. 4d. Broad peaks indicative of a short-range order (SRO) are observed above $T_N$, and additional sharp Bragg peaks from the LRO appear below $T_N$, with the broad peaks almost unchanged. The large intensity of the broad peaks relative to those of the Bragg peaks indicates the presence of strong spin fluctuations in pharmacosiderite even in the LRO state.

A striking feature of the broad peaks is their asymmetric shape. Such a peak shape with a long tail toward high $Q$ is generally known to be a characteristic of the powder diffraction profile from low-dimensional spin systems. In this study, we analysed the paramagnetic scattering of pharmacosiderite at 7.2 K in terms of the low-dimensional scattering model[19,20] (Supplementary Note 7). As shown in Fig. 4e, the 2D model best reproduces the

experimental spectrum with a coherence length within the plane $2\pi D^{-1}$ estimated to be 9 times the lattice constant, ~70 Å. Thus, a 2D SRO exists in pharmacosiderite despite the cubic symmetry of the underlying lattice. Because similar broad peaks remain below $T_N$, this 2D SRO must survive as a large 2D spin fluctuation in the LRO.

By employing the linear SW theory for the Heisenberg spin model, we have examined whether such a 2D spin fluctuation is, in fact, possible in pharmacosiderite. Figure 4f shows the calculated SW dispersions for the $\mathbf{q} = 0$, $\Gamma_5$ magnetic structure. Four SW modes appear as there are four sites in the unit cell. Notably, all of them are completely flat at zero energy along Γ–Z without DM interactions (solid lines in Fig. 4f) or only weakly dispersive at finite energies with DM interactions (dash-dotted lines), while being highly dispersive along most other directions. This strongly suggests that the motion of magnons tends to be confined within the $c$ layer in the low-energy region, leading to two-dimensionality in the spin excitation despite the isotropic magnetic couplings.

**Two-dimensionalisation by frustration**. Here, we discuss the mechanism leading to the observed two-dimensionality in pharmacosiderite within the classical Heisenberg spin model. Figure 5a illustrates the intercluster couplings in the $\mathbf{q} = 0$ and $\Gamma_5$ structures with the $c$ plane as a coplanar plane. A spin in a tetrahedron interacts with three pairs of antiferromagnetically aligned spins in the nearby tetrahedra via the identical Heisenberg interaction $J'$. Notably, towards the pair along the $c$ axis, the two $J'$ paths (sky-blue bonds in Fig. 5a) are geometrically cancelled, whereas the $J'$ paths (orange bonds) toward either pair along the $a$ or $b$ axis do not cause such a cancellation. As a result, the 3D network is transformed into an array of decoupled layers with the remaining in-plane interactions of the order of $J'$. The 2D feature in the $S(Q, E)$s observed in our experiments, and the SW calculations may be attributed to this dimensional reduction or two-dimensionalisation by geometrical frustration. We emphasise that, compared to the ATL or BCT antiferromagnets (Fig. 1), the resulting 2D anisotropy is not a feature of the original crystal lattice but is induced by the evolution of AF correlations. Because the unique axis perpendicular to the layers is not fixed to a certain crystallographic direction, it should be equally chosen among all three identical [100] directions. Further, it can be controlled by external magnetic fields, as will be mentioned later.

In an actual crystal of pharmacosiderite, we must consider a modification arising from small anisotropic interactions in addition to the major Heisenberg interactions. The geometrical cancellation in the $J'$ couplings along the $c$ axis should become imperfect owing to the DM interaction and the spin canting by the single-ion anisotropy, as reflected by the appearance of the weak dispersions along Γ–Z in the SW calculations (Fig. 4f). Nevertheless, as these anisotropy terms are quite small, to preserve a certain flatness in the Γ–Z modes, the influence may be negligible except at sufficiently low temperatures; judging from the dispersions in 0–0.2 meV, two-dimensionalisation likely occurs above ~2 K.

We have demonstrated that 2D spin fluctuations exist in the $\mathbf{q} = 0$, $\Gamma_5$ order of pharmacosiderite. In general, however, such a 2D fluctuation can reduce the ordered magnetic moment slightly, but not that much. In fact, our SW calculations with the anisotropies found a reduction to 4.0 $\mu_B$ at 0 K, which is still much larger than the experimental value of 2.1 $\mu_B$. However, this calculated value is close to the maximum static magnetic moment estimated by Mössbauer spectroscopy. This observation means that the local magnetic moment itself is not reduced by quantum fluctuations, whereas the average one over the LRO is likely reduced by certain spatial and temporal fluctuations.

**One-dimensionalisation generating 1D defects**. The origin of the large reduction in the ordered moment may be ascribed to the 1D spin fluctuations generated in the 2D layer of the $\mathbf{q} = 0$, $\Gamma_5$ order. Notably, in the SW calculations of Fig. 4f without anisotropies, there is a zero-energy, flat excitation mode along Γ–Z–R–X–Γ (red curve), in addition to those along Γ–Z mentioned above. This mode is not a local zero-dimensional mode because it exhibits a large dispersion along Γ–Z. Instead, it may be a 1D mode occurring along the $a$ axis because its dispersion is flat

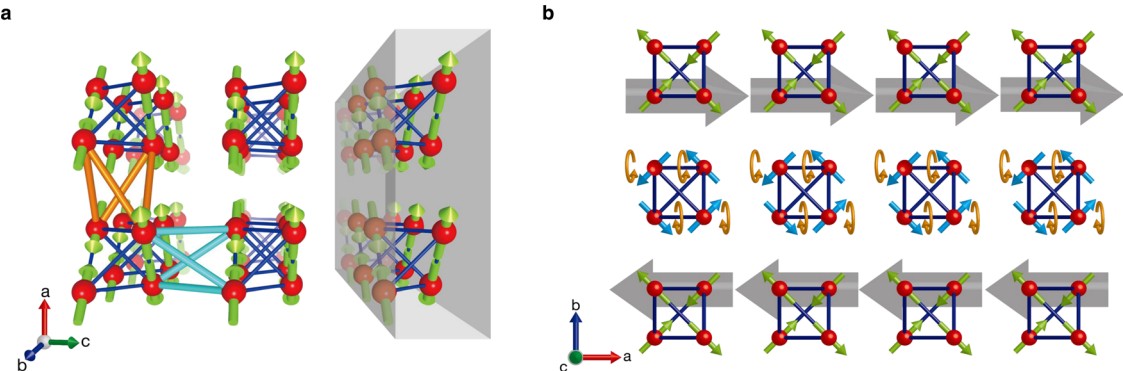

**Fig. 5 Two- and one-dimensionalisation by geometrical frustration in pharmacosiderite. a** Schematic representation of the $\mathbf{q} = 0$, $\Gamma_5$ magnetic structure with all the spins perpendicular to the horizontal $c$ axis. The red, blue, and green arrows indicate the crystallographic $a$, $b$, and $c$ axes, respectively. Among the six intercluster $J'$ paths from a focused spin in a tetrahedron, the two paths to the pair of antiparallel spins along the $c$ axis (sky-blue bonds) geometrically cancel each other because the total $S$ equals 0 for the pair. In contrast, the paths to either pair of spins along the $a$ or $b$ axis (orange bonds) do not; spins in either pair are not antiparallel to each other. This results in a slab perpendicular to the $c$ axis, in which SWs are effectively confined. **b** 1D defect along the $a$ axis, which is made of $\Gamma_4$ tetrahedra with blue spins and is generated without energy loss in the slab shown in **a** of the $\mathbf{q} = 0$ magnetic structure made of $\Gamma_5$ tetrahedra with green spins. The red, blue, and green arrows indicate the crystallographic $a$, $b$ and $c$ axes, respectively. The large grey arrow represents the sum of a pair of spins at an edge of the tetrahedron next to the 1D defect, which is parallel to the $a$ axis and exerts an effective axial magnetic field to the spins of the 1D defect. Under the axial effective fields, all the blue $\Gamma_4$ spins in the 1D defect can rotate collectively along the $a$ axis while keeping the total spin zero and are continuously transformed into the $\Gamma_5$ arrangement without losing energy in the intercluster couplings to the surrounding. This means that such a 1D defect can be generated and annihilated without energy loss in the 2D slab of the $\mathbf{q} = 0$, $\Gamma_5$ magnetic structure in the classical spin model.

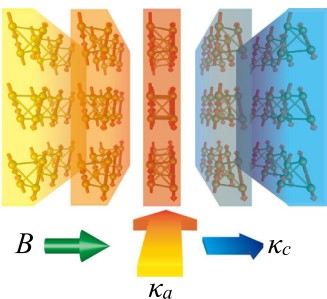

$B \Rightarrow$ $\Leftarrow \kappa_c$

$\kappa_a$

**Fig. 6 Concept of magnetic-field control of thermal conductivity.** When magnetic-field $B$ is applied horizontally along one of the cubic axes of pharmacosiderite in its $\mathbf{q} = 0$, $\Gamma_5$ magnetic order, magnon excitations are confined in each vertically aligned layer; this selection of the magnetic $c$ axis occurs because the parasitic ferromagnetic moment along the $c$ axis prefers the field direction even if $B$ is as small as 0.5 T. Because magnons can carry a heat flow, the thermal conductivities $\kappa_a$ and $\kappa_c$ become large and small along directions perpendicular and parallel to $B$, respectively, which results in a large horizontal temperature gradient, as depicted by the variation in colour. When $B$ is rotated to the vertical direction, the anisotropic thermal conduction is reversed. Thus, a switching of thermal conductance becomes possible via the external magnetic field.

along the $b^*c^*$ plane. We plausibly assume that the 1D mode originates from a line defect in the $\Gamma_5$ layer as a result of geometrical cancellation of intercluster interactions in the isotropic spin model. As depicted in Fig. 5b and described in detail in Supplementary Note 8, all the spins in a string of clusters along the $a$ axis can rotate collectively so as to keep all the intercluster couplings to the surroundings always AF. As a result, there is no energy cost to generate such a 1D defect made of $\Gamma_4$ clusters in the $\mathbf{q} = 0$, $\Gamma_5$ order as a zero-energy mode. Therefore, one-dimensionalisation by frustration should take place together with two-dimensionalisation in this unique spin system. More solid experimental evidence may require further inelastic neutron scattering studies using a single crystal of pharmacosiderite.

True zero-energy excitation modes should destroy the LRO completely. However, the order-by-disorder effect may lift the degeneracy in the presence of thermal or quantum fluctuation[21,22]. In the present compound, the small anisotropies also lift it to have a finite energy gap; this is not exactly a true energy gap but a pseudo gap with a few dispersive modes existing below the flat modes in the SW calculation in Fig. 4f. Thus, they are scarcely populated at $T = 0$. Nevertheless, at high temperatures, many of them can be thermally activated and partly destroy the LRO, which may produce a large 1D spin fluctuation that causes a reduction in the average ordered moment in the LRO. This may be the reason for the observed large reduction in the ordered moment and the coexistence of the static and dynamic magnetisms probed by Mössbauer and μSR experiments (Supplementary Notes 2, 3).

**Possible thermal conductivity switching.** Finally, we propose a concept based on the present findings in pharmacosiderite. It is well established that an array of spins can carry a heat flow in addition to the lattice[23]. Specifically, in 1D spin systems, the thermal conductivity is notably enhanced along the chains[24,25]. Spin contributions to thermal conductivity can be larger than lattice contributions in some frustrated magnets[26–28]. For pharmacosiderite, one expects a larger thermal conductivity along the $c$ layer as magnon excitations are confined in the layer. Interestingly, the direction of the $c$-axis can be controlled by an external magnetic field because it is not fixed to the crystallographic lattice. Rather, it is selected by the direction of the weak

ferromagnetic moment of the magnetic order; even a small magnetic field of 0.5 T can control it[16]. Thus, the thermal conductivity along any direction is enhanced and reduced when the field is perpendicular and parallel to the direction, respectively, as schematically illustrated in Fig. 6. Notably, a difference in thermal conductivity along the two directions only measures the spin contributions. Therefore, magnetic-field switching of thermal conductivity might be possible in pharmacosiderite. Future thermal conductivity measurements using a single crystal or even a polycrystalline sample would confirm this intriguing idea.

## Methods

**Sample preparation.** A powder sample of pharmacosiderite was synthesised by a conventional hydrothermal method[16]. First, a beige colloidal precursor was obtained by vigorous stirring of a ferric solution (30 g of $NH_4Fe(SO_4)_2 \cdot 12H_2O$ fully dissolved in 7.5 g of water) and an arsenate solution (8 g of $KH_2AsO_4$ and 7.5 g of $K_2CO_3$ fully dissolved in 19 g of water). After the pH of the precursor was adjusted to approximately 1.5 by the addition of 0.1 mL of 10 M HClaq, the precursor was heated in a Teflon-lined autoclave for three h at 220 °C. After the reaction, a pale-yellow precipitate of potassium pharmacosiderite $KFe_4(AsO_4)_3(OH)_4 \cdot nH_2O$ ($n = 8-9$) was filtered and thoroughly washed with water. Subsequently, the obtained powder was annealed in 500 mL of 0.1 M HClaq for a week at 100 °C. Finally, hydronium pharmacosiderite $(H_3O)Fe_4(AsO_4)_3(OH)_4 \cdot 5.5H_2O$ was obtained as a pale-green powder after filtration and drying. Deuterated ingredients were used for the neutron scattering experiments. Cation and deuterium substitution have negligible effects on the magnetic properties. Single crystals of pharmacosiderite were synthesised under hydrothermal conditions under a temperature gradient[29]. A quartz ampoule with a length of 150 mm was filled with 0.1 g of polycrystalline pharmacosiderite and 5 mL of solution remained after powder synthesis. A thick quartz tube with outer and inner diameters of 12 and 8 mm, respectively, was used to avoid explosion. The ampoule was placed vertically in a two-zone furnace and heated at 200 °C at the hot bottom end and 120 °C at the cold top end for a month. Seething of the solution caused polycrystalline pharmacosiderite to decompose and recrystallise near the liquid level. This resulted in the growth of cube-shaped crystals as large as $0.3 \times 0.3 \times 0.3$ mm.

**Magnetisation measurements.** Magnetisation measurements were conducted using a single crystal of pharmacosiderite in a magnetic property measurement system 3 (MPMS-3, Quantum Design).

**Heat capacity measurements.** Heat capacity measurements were conducted using powder pharmacosiderite in a physical property measurement system (Quantum Design).

**Muon spin rotation/relaxation measurements.** Conventional μSR experiments were conducted on two types of pharmacosiderite: $AFe_4(AsO_4)_3(OH)_4 \cdot nH_2O$ with $A = K$ and $H_3O$. The K sample was examined in an ARTEMIS spectrometer in the S1 area at the J-PARC MUSE. A 100% spin-polarised pulsed beam of positive muons with a momentum of 27 MeV/c and a full-width-at-half-maximum of $t_{PW} \simeq 100$ ns was delivered to a powder sample loaded on a He-flow cryostat. Time-dependent μ–e decay asymmetry $A(t)$ was measured under a zero field (ZF), a longitudinal field (LF), or a weak transverse field (TF) over a temperature range from 4 to 300 K. Additional measurements for the $H_3O$ sample were conducted in a Lampf spectrometer on the M20 beamline at TRIUMF, Canada, to observe a fast depolarisation emerging at 2 K in the early time range of $A(t)$ with a higher time resolution of ~1 ns.

**Neutron diffraction experiments.** Powder neutron diffraction experiments were performed using constant-wavelength ($\lambda = 2.4395$ Å) diffractometer ECHIDNA in ANSTO at 1.6 and 10 K, which are above and below $T_N = 6$ K. Rietveld refinements were performed to determine the magnetic structure using the programme Fullprof Suite[30]. Crystal and magnetic structures were drawn using VESTA[31].

**Inelastic neutron scattering experiments.** Inelastic neutron scattering experiments were performed in time-of-flight spectrometer AMATERAS[32] at J-PARC at 0.6, 1.6, 2.5, 3.5, 5, 7.2, 8, 20 and 50 K at incident energies of 3.1 and 7.7 meV, where the energy resolutions at the elastic position were 0.058 and 0.43 meV (full width at half maximum), respectively. The raw data were reduced by the UTSU-SEMI software suite[33].

**Spin-wave calculations.** Linear SW analysis was conducted using the SpinW software[34].

## Data availability

The publication data used in this study is available at https://doi.org/10.6084/m9.figshare.13996436.v1. The data files are also available from the corresponding author upon request.

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

## Acknowledgements

We are grateful to Gøran. J. Nilsen, H. Yan, H. Tsunetsugu, M. Ogata, N. Shannon, S. Hayashida, Y. Kato and Y. Motome for helpful discussions. We thank Kazuaki Iwasa for providing us with a cryostat for inelastic neutron scattering experiments. R.O. is supported by the Materials Education Programme for the Future Leaders in Research, Industry, and Technology (MERIT) given by the Ministry of Education, Culture, Sports, Science and Technology of Japan (MEXT). This work was partially supported by KAKENHI (Grant No. 15K17701) and the Core-to-Core Programme for Advanced Research Networks given by the Japan Society for the Promotion of Science (JSPS). The neutron diffraction experiment at ECHIDNA was supported by a General User Program for Neutron Scattering Experiments, Institute for Solid State Physics, The University of Tokyo (Proposal No. 17401 and No. 17403). The neutron and muon experiment at the Materials and Life Science Experimental Facility of the J-PARC was performed under approved proposals No. 2016MI21, 2017I0014, and 2018I0014.

## Author contributions

R.O., M.K. and Z.H. conceived and designed the study. R.O., M.K. and K.N. performed inelastic neutron scattering experiments. S.A., M.A. and T.M. performed the diffraction experiments. S.A. analysed the diffraction data. A.K., H.O., M.H., S.T., K.M.K. and R.K. performed the μSR experiments. A.K. analysed the μSR data. R.O., M.K. and Z.H. interpreted all the experimental data. R.O. performed linear SW calculations. The paper was written based on the discussion by all authors.

## Competing interests

The authors declare no competing interests.
