## [Peer Review File · Nature Communications]

REVIEWER COMMENTS

Reviewer #1 (Remarks to the Author):

The authors report an interesting case of dimensional reduction induced by geometric frustration. More specifically, the classical ground state of certain frustrated magnets can exhibit a lower effective dimensionality because the projected Hamiltonian has symmetries that are not present in the rest of the spectrum (see for instance PRB 77, 094406 (2008)). As the authors discuss in their manuscript, this spontaneous reduction of the dimensionality leads to a dynamically induced anisotropy that can be exploited to achieve magnetic control of the conductivity tensor.

The material under consideration is the tetrahedral cluster antiferromagnet pharmacosiderite that comprises spin $5/2$ Fe^{3+} ions. The authors combine muon spin resonance and powder-averaged inelastic neutron scattering measurements to reveal the presence of one and two-dimensional fluctuations that manifest via the corresponding zero modes of the magnon spectrum. These zero modes are removed by small anisotropic interactions that introduce a hierarchy of small energy scales, which is necessary to explain the observed sequence of dimensional reductions: 3D-2D at a higher energy and 2D-1D at lower energies.

The experimental data is of very high quality and the data analysis is scientifically sound. The direct observation of dimensional reduction in a cubic material is indeed very interesting because, as the authors suggest by the end of the manuscript, it opens the door for external control of the spontaneous anisotropy. In my opinion, this manuscript can be suitable for publication in Nature Communications after the authors consider and address the following points:

-The English should be polished in several sentences of the manuscript. For instance, “reveal a persistent fluctuations” should be replaced by “reveal persistent fluctuations”

-The poor agreement between LSW theory (Fig. 4c) and the experiment can be attributed to the fact that the linear spin wave theory corresponds to an expansion around a particular classical ground state of the system. I wonder if the authors considered the possibility of using classical Monte Carlo simulations plus Landau Lifshitz dynamics to account for the ground state degeneracy and the observed dimensional reduction.

-The moment reduction is attributed to the 2D-1D dimensional reduction and the consequent large thermal fluctuations of the ordered moment. Is it clear from some aspect of the data that the moment reduction is cannot be attributed to disorder or structural defects?

-Although it does not seem to play a role here because of the presence of small interactions that remove the zero modes, the authors may want to mention that order by disorder (mostly induced by thermal fluctuations for a spin 5/2 system) can also lift the degeneracy associated with the zero modes and induce a particular 3D ordering at finite T.

-I have a couple of questions about the magnetic control of the thermal conductivity tensor. The first one is: what is the magnon contribution to the thermal conductivity relative to the lattice contribution in pharmacosiderite? The second questions is: what is the external field required to remove the domains oriented in the different directions, i.e. the critical field that is necessary to exchanges the principal directions of the conductivity tensor? I am asking this question because the ferromagnetic moment is weak. I suggest to include some of this information in the last section of the manuscript.

Reviewer #2 (Remarks to the Author):

This paper introduces an new frustrated spin system with interesting analogies to the famous spin ice materials. The cubic unit cell consists of a single tetrahedron and this leads to intriguing degeneracies and apparently suppression of the ordering temperature and magnetic moment size. The model has not been widely considered before and I think the paper would spark considerable theoretical interest in what forms of quantum magnetism might be possible on this lattice as well as efforts to find other experimental realizations.

This is a comprehensive paper including multiple experimental studies and efforts to model the data with spin wave theory. I do recommend publication of the paper in Nature communications after the following comments are addressed preferably in the text of the paper. The main theme is to clearly acknowledge and discuss the potential for structural defects and their impacts on the magnetism.

1. what is the correlation length of the magnetic ordered state. The powder diffraction data can provide a lower limit and perhaps also an upper limit.

2. What is the time scale that the Moessbauer data sets on any temporal spin fluctuations that might lead to the broadening of the absorption lines. Could the broadening not be associated with static disorder.

3. In the supplemental information it is mentioned that there is "...large positional disorder of crystal waters." It should be clearly stated whether this is disorder of the water minority phase or the D2O molecules within the magnetic material. The volume fraction of the D2O minority phase should be provided.

4. There should be a discussion of the potential impact of structural disorder on the magnetic interactions within and between tetrahedra. This may be essential for understanding the magnetism and the relevant information should come out clearly in the presentation so theorists and experimentalist can consider it and make progress.

5. The fact that the muon is not probing the magnetism of the equilibrium crystal structure but the magnetism in the presence of a charged defect (the muon) should be made clear in the presentation. This could be important for interpreting the data.

Reviewer #3 (Remarks to the Author):

The authors present a very detailed and interesting magnetic study of the pharmacosiderite compound characterized by competing interactions. This is follow-up of ref.15 by Okuma et al. using neutron and muon experiments.

Due to my field of (in-)expertise, I will not comment on the details of the analysis of elastic and inelastic neutron studies which appear as the central part of the manuscript. I hope another referee does!

The magnetic structure and spin waves of this compound are revealed in this study. The work was performed along good standards, including the deduction of anisotropies from a spin wave analysis and the structure from neutron refinements. One of the strongest argument embedded in the analysis of the neutron data is that the small value of the moment detected through neutron measurements can be associated with a dimensional reduction down to 1D. At this stage of the review, I am not convinced, see below.

1- There seems to be a severe contradiction (p.6 of the manuscript) between the neutron analysis pointing to a 1.9 μ_B moment and the value given by Mössbauer experiment of 4.0 μ_B . Is there a problem with the neutron analysis?

This reduction of the moment copes with the idea of a muon evidence for persistent fluctuations in the ordered phase.

2-Yet, I have some concerns with the very involved analysis of the muon data detailed in the supplementary material where the authors argue that there are two very different sites, one which witnesses the magnetic ordering and the other one which senses non-frozen fluctuating moments which are speculatively associated with fluctuating moment due to 1D clusters.

Can the authors discard a simpler one-site analysis where only one species of frozen moment?

(a) from the 59 K asymmetry, one could in principle deduce the background, if any. I recognize that it might be difficult to introduce a non-relaxing component while already using two different fractions corresponding to two different sites – two sites sounds wise to me. What is the f value?

I am also somehow puzzled by the three spins analysis of the oscillations. From the structure presented in fig. S2, I cannot figure out whether there are two proximate OH ions (3 spins relax.) or one (two spins relaxation). If not, then the OH- μ asymmetry would also display oscillations but rather due to a two spins model [see Lord et al., *Physica B* 289-280, 495c (2000) and Mendels et al., *Phys. Rev. Lett.* 98, 077204 (2007)]. Isn't the OH- μ distance deduced from the 3 spins analysis somehow large for a μ -OH- μ complex. Wouldn't a two spins, ie OH- μ complex analysis yield a better value of the distance. A proper analysis might change the f fraction and possibly the background, which impacts the discussion of the following item.

(b) the static part should have a 1/3rd tail –this would be true in the case of two muons sites both feeling frozen moments. one expects that this 1/3rd tail is decoupled with the applied field, and its time dependent part is associated with relaxation. The estimate of the background at high-T enables one to assess whether all the signal detected is in the 1/3rd tail or not. If all the relaxation would be in the 1/3rd tail, then there is no need of the involved analysis performed by the authors, ie the assignment to specific sites is not granted especially a muon site coupled in isolation to the 1D excitation modes

3- The authors should also use their Mossbauer low-T spectrum to determine at least an order of magnitude the frequency of the slow spin fluctuations, see e.g. Bonville et al. *PRL* 92 (2004) 167202 or *Phys. Rev. Lett.* 88, 077204 (2002).

Because of these reservations, I feel uncomfortable with the idea of the “1D” dimensional reduction. I feel this has to be clarified before the work can be published.

We provide point-by-point responses to the referees' comments below. Referees' and our comments are shown in black and blue, respectively. Changes in the revised manuscript and Supplementary Information are shown in red.

Reviewer #1 (Remarks to the Author):

The authors report an interesting case of dimensional reduction induced by geometric frustration. More specifically, the classical ground state of certain frustrated magnets can exhibit a lower effective dimensionality because the projected Hamiltonian has symmetries that are not present in the rest of the spectrum (see for instance PRB 77, 094406 (2008)). As the authors discuss in their manuscript, this spontaneous reduction of the dimensionality leads to a dynamically induced anisotropy that can be exploited to achieve magnetic control of the conductivity tensor.

Thank you for the note. We have cited this paper in the third paragraph, referring to $\text{BaCu}_2\text{Si}_2\text{O}_6$ on page 1.

The material under consideration is the tetrahedral cluster antiferromagnet pharmacosiderite that comprises spin $5/2$ Fe^{3+} ions. The authors combine muon spin resonance and powder-averaged inelastic neutron scattering measurements to reveal the presence of one and two-dimensional fluctuations that manifest via the corresponding zero modes of the magnon spectrum. These zero modes are removed by small anisotropic interactions that introduce a hierarchy of small energy scales, which is necessary to explain the observed sequence of dimensional reductions: 3D-2D at a higher energy and 2D-1D at lower energies.

The experimental data is of very high quality and the data analysis is scientifically sound. The direct observation of dimensional reduction in a cubic material is indeed very interesting because, as the authors suggest by the end of the manuscript, it opens the door for external control of the spontaneous anisotropy. In my opinion, this manuscript can be suitable for publication in Nature Communications after the authors consider and address the following points:

We thank the referee for appreciating the novelty of our work.

-The English should be polished in several sentences of the manuscript. For instance, “reveal a persistent fluctuations” should be replaced by “reveal persistent fluctuations”

We are sorry for our poor English writing. We have corrected these in the revised manuscript with the help of a proofreading service.

-The poor agreement between LSW theory (Fig. 4c) and the experiment can be attributed to the fact that the linear spin wave theory corresponds to an expansion around a particular classical ground state of the system. I wonder if the authors considered the possibility of using classical Monte Carlo simulations plus Landau Lifshitz dynamics to account for the ground state degeneracy and the observed dimensional reduction.

We performed neither classical Monte Carlo nor Landau Lifshitz simulations. This is because the LSW theory has already captured several key features of the excitation, such as two-dimensionality, band width, and the size of the anisotropy gap, which supports the validity of the spin Hamiltonian and the dimensional reduction in pharmacosiderite. The experimental spectrum shown in Fig. 4a, however, is much broader than the calculation based on the LSW theory in Fig. 4c. For further quantitative evaluations of the dynamics, more elaborate analyses are necessary and will be considered in the future. We agree that Monte Carlo or Landau Lifshitz calculations can reproduce the broadness of the spectrum and the size of the ordered moment.

-The moment reduction is attributed to the 2D-1D dimensional reduction and the consequent large thermal fluctuations of the ordered moment. Is it clear from some aspect of the data that the moment reduction is cannot be attributed to disorder or structural defects?

Structural disorder may play some role in moment reduction, although it is difficult to estimate the contribution quantitatively. However, we point out that structural disorders such as site-mixing or positional disorder for pharmacosiderite should produce isotropic diffuse scattering instead of the observed dominant low-dimensional scattering. Thus, we believe that 2D-1D dimensional reduction is the predominant cause of the moment reduction. Consequently, we have modified the sentences in the second paragraph of 'Novel 3D cluster magnet pharmacosiderite' on page 2.

-Although it does not seem to play a role here because of the presence of small interactions that remove the zero modes, the authors may want to mention that order by disorder (mostly induced by thermal fluctuations for a spin 5/2 system) can also lift the degeneracy associated with the zero modes and induce a particular 3D ordering at finite T.

We considered the Dzyaloshinskii–Moriya interaction and single ion anisotropy as the origin of the finite temperature transition. However, it is true that order by disorder can cause an ordering. We have added a related sentence to the second paragraph of ‘One-dimensionalisation generating 1D defects’ in page 5.

-I have a couple of questions about the magnetic control of the thermal conductivity tensor. The first one is: what is

the magnon contribution to the thermal conductivity relative to the lattice contribution in pharmacosiderite? The second question is: what is the external field required to remove the domains oriented in the different directions, i.e. the critical field that is necessary to exchange the principal directions of the conductivity tensor? I am asking this question because the ferromagnetic moment is weak. I suggest to include some of this information in the last section of the manuscript.

We do not have information on how much magnons contribute to the thermal conductivity in pharmacosiderite. Closely related examples are found in kagome minerals; spin contributions below the Neel temperature vary from 20 % in Ca-kapellasite (*PRL* 121, 097203 (2018)) to 80% in Cd-kapellasite (*PRX* 10, 041059 (2020)). We would like to emphasise that only the magnon contribution should change with the applied field direction so that it can be estimated via future thermal conductivity measurements.

The magnetic field necessary for aligning magnetic domains is rather small—in the order of the coercive field of 0.5 T at 2 K (ref. 16), even though the ferromagnetic moment is small. This magnetic field can be produced by a permanent magnet, such as a neodymium magnet. We have addressed these points in the last paragraph of the revised manuscript.

Reviewer #2 (Remarks to the Author):

This paper introduces a new frustrated spin system with interesting analogies to the famous spin ice materials. The cubic unit cell consists of a single tetrahedron and this leads to intriguing degeneracies and apparently suppression of the ordering temperature and magnetic moment size. The model has not been widely considered before and I think the paper would spark considerable theoretical interest in what forms of quantum magnetism might be possible on this lattice as well as efforts to find other experimental realizations.

This is a comprehensive paper including multiple experimental studies and efforts to model the data with spin wave theory. I do recommend publication of the paper in Nature communications after the following comments are addressed preferably in the text of the paper. The main theme is to clearly acknowledge and discuss the potential for structural defects and their impacts on the magnetism.

We appreciate the referee's recommendation for publication.

1. what is the correlation length of the magnetic ordered state. The powder diffraction data can provide a lower limit and perhaps also an upper limit.

As shown in the following figures, the nuclear and magnetic peaks at the (001) reflection have similar widths, indicating that the magnetic order occurs over a long range with the magnetic correlation length being limited by the crystallite domains. A microstructural analysis performed by Fullprof yielded a magnetic correlation length of 360.7(3) Å, which is 45 times longer than the length of the axis. We have added a few sentences in the third

paragraph of 'Novel 3D cluster magnet pharmacosiderite' on page 2 and in Section 4 of the Supplementary Information.

2. What is the time scale that the Moessbauer data sets on any temporal spin fluctuations that might lead to the broadening of the absorption lines. Could the broadening not be associated with static disorder.

To answer this question, we have carried out a new analysis of the Mössbauer data based on the Blume–Tjon model, which takes into account the stochastic fluctuation of Ising-like spins. By fitting the spectra, the average frequency of the fluctuation was determined to be 313(72) MHz at 2.8 K. This time scale is of the same order as that deduced from the μ SR experiments, that is, 427(10) MHz at 1.6 K. Therefore, we attribute the broadening of the Mössbauer spectra to temporal fluctuations rather than to the static distribution of hyperfine fields due to crystalline disorder. We have added corresponding descriptions in the second paragraph of 'Novel 3D cluster magnet pharmacosiderite' and Supplementary Section 2 in the revised manuscript.

3. In the supplemental information it is mentioned that there is "...large positional disorder of crystal waters." It should be clearly stated whether this is disorder of the water minority phase or the D2O molecules within the magnetic material. The volume fraction of the D2O minority phase should be provided.

There was no water-minority phase. We considered only pharmacosiderite with a fixed water content and a positional disorder of D₂O. For clarification, a description has been added to Section 1 of the Supplementary Information.

4. There should be a discussion of the potential impact of structural disorder on the magnetic interactions within and between tetrahedra. This may be essential for understanding the magnetism and the relevant information should come out clearly in the presentation so theorists and experimentalist can consider it and make progress.

The structural disorder in pharmacosiderite results from two types of water molecules and hydronium ions incorporated into the open framework structure. In the revised manuscript, we carefully dealt with the disorder to obtain a realistic model. The crystal water (O3-D1) near the face centre of the primitive cubic cell has only an orientational disorder and takes one of two orientations. This may have little influence on the exchange interaction because it is isolated from the Fe network, as depicted in Fig. S1c. On the other hand, the O4-D3 unit (Fig. S1d) contains both orientational and occupational disorders; it may exist as a water molecule or a hydronium ion, or the

site may be absent. This forms a hydrogen bond with the hydroxide ligand (O2-D2) of Fe³⁺ and may be more influential to the intracluster interaction J rather than to the intercluster interaction J' . However, it is difficult to know how much this disorder affects the magnetic ground state and fluctuations. We described these types of disorders of the crystal waters in the refined structure (Supplementary 1, Figs. S1c and S1d).

5. The fact that the muon is not probing the magnetism of the equilibrium crystal structure but the magnetism in the presence of a charged defect (the muon) should be made clear in the presentation. This could be important for interpreting the data.

We indeed observed an oscillatory signal that evidences the formation of a charged “nonequilibrium” structure above the Néel temperature. We considered this effect and considered two structural models of the local cluster of muon and hydrogen. Then, we analysed the magnetic fluctuations in both cases and found that all our interpretations are consistent with each other. We would like to stress that the analysis on the Mossbauer data suggests a dynamically fluctuating magnetism at a frequency of 310(72) MHz, which is in good agreement with the results obtained by muon experiments. The details are described in the revised Supplementary Section3.

Reviewer #3 (Remarks to the Author):

The authors present a very detailed and interesting magnetic study of the pharmacosiderite compound characterized by competing interactions. This is follow-up of ref.15 by Okuma et al. using neutron and muon experiments.

Due to my field of (in-)expertise, I will not comment on the details of the analysis of elastic and inelastic neutron studies which appear as the central part of the manuscript. I hope another referee does!

The magnetic structure and spin waves of this compound are revealed in this study. The work was performed along good standards, including the deduction of anisotropies from a spin wave analysis and the structure from neutron refinements. One of the strongest argument embedded in the analysis of the neutron data is that the small value of the moment detected through neutron measurements can be associated with a dimensional reduction down to 1D. At this stage of the review, I am not convinced, see below.

1- There seems to be a severe contradiction (p.6 of the manuscript) between the neutron analysis pointing to a 1.9 μ_B moment and the value given by Mössbauer experiment of 4.0 μ_B . Is there a problem with the neutron analysis?

We believe that the two results are consistent with each other. According to our model and assuming that one-dimensional defects disturb the Γ_5 order, the length of each spin is close to 4 μ_B , as expected from the linear spin wave analysis, whereas a substantial number of spin clusters show a deviation from the perfect Γ_5 structure. As Mössbauer spectroscopy is a local probe, it can directly measure the size of the local spin moment, whereas the ordered moment from neutron diffraction should be reduced depending on the density of one-dimensional defects because neutron

diffraction measures the degree of periodicity. This point was mentioned in the second paragraph of 'Two-dimensionalisation by frustration' in the original text.

This reduction of the moment copes with the idea of a muon evidence for persistent fluctuations in the ordered phase.

2-Yet, I have some concerns with the very involved analysis of the muon data detailed in the supplementary material where the authors argue that there are two very different sites, one which witnesses the magnetic ordering and the other one which senses non-frozen fluctuating moments which are speculatively associated with fluctuating moment due to 1D clusters.

Can the authors discard a simpler one-site analysis where only one species of frozen moment?

As mentioned below, the analysis assuming two relaxing components consistently gives the best fitting results both above and below the Néel temperature.

(a) from the 59 K asymmetry, one could in principle deduce the background, if any. I recognize that it might be difficult to introduce a non-relaxing component while already using two different fractions corresponding to two different sites – two sites sounds wise to me. What is the f value?

I am also somehow puzzled by the three spins analysis of the oscillations. From the structure presented in fig. S2, I cannot figure out whether there are two proximate OH ions (3 spins relax.) or one (two spins relaxation). If not, then the OH- μ asymmetry would also display oscillations but rather due to a two spins model [see Lord et al., Physica B 289-280, 495c (2000) and Mendels et al., Phys. Rev. Lett. 98, 077204 (2007)]. Isn't the OH- μ distance deduced from the 3 spins analysis somehow large for a μ -OH- μ complex. Wouldn't a two spins, ie OH- μ complex analysis yield a better value of the distance. A proper analysis might change the f fraction and possibly the background, which impacts the discussion of the following item.

We analysed the μ SR data for both the H- μ -H model and the H- μ model, the results of which are shown in Figs. S3a and S3b in the revised Supplementary Information, respectively. Our discussion in the Supplementary Information is based on the H- μ -H model because it gives a slightly better χ^2/N value than the H- μ model. However, there was no remarkable difference between the analyses that would require a major change in the discussion, as the Referee mentioned. The constant background turned out to be negligible in both models, and the f -values were similar: 0.21 for the H- μ -H model and 0.30 for the H- μ model. The distances between H and μ were estimated to be 1.8 and 1.55 Å for the H- μ -H and H- μ models, respectively. The distance between neighbouring hydrogens (D2 in Fig. S1c) surrounding the Fe tetrahedron was 4.1 Å. Suppose that the muon stops at the centre of a straight line connecting these hydrogens and the estimated distance between H and μ is 2.1 Å. Thus, the H- μ -H model is more appropriate than the H- μ model.

(b) the static part should have a 1/3rd tail –this would be true in the case of two muons sites both feeling frozen moments. one expects that this 1/3rd tail is decoupled with the applied field, and its time dependent part is associated with relaxation. The estimate of the background at high-T enables one to assess whether all the signal detected is in the 1/3rd

tail or not. If all the relaxation would be in the 1/3rd tail, then there is no need of the involved analysis performed by the authors, ie the assignment to specific sites is not granted especially a muon site coupled in isolation to the 1D excitation modes

As mentioned in the previous answer, the background was found to be negligibly small at $T > T_N$. Therefore, below T_N , the component that does not show relaxation, as seen in Fig. S3c in the revised Supplementary Information, strongly suggests that there is a 1/3 component due to the static internal magnetic field and that the muon at the $\mu(2)$ -site simultaneously undergoes dynamic fluctuation.

3- The authors should also use their Mossbauer low-T spectrum to determine at least an order of magnitude the frequency of the slow spin fluctuations, see e.g. Bonville et al. PRL 92 (2004) 167202 or Phys. Rev. Lett. 88, 077204 (2002).

Thank you for providing us this helpful information about how to estimate the energy scale in Mössbauer spectroscopy. We analysed the Mössbauer spectroscopy data using a simple stochastic model and obtained a frequency of 310(72) MHz, which is similar to that deduced from the μ SR data. The details of the analysis are given in the revised Supplementary Information.

Because of these reservations, I feel uncomfortable with the idea of the “1D” dimensional reduction. I feel this has to be clarified before the work can be published.

In the revised manuscript, we have shown that the results of Mössbauer spectroscopy and muon experiments are consistent with each other and clearly point to the existence of low-energy fluctuation. The low-dimensional character of the excitation is already evidenced by neutron scattering. Thus, it is likely that the one-dimensional defects expected for the realised magnetic structure explain all of these features. We hope that the referee is now convinced by the idea of one-dimensionalisation.

REVIEWERS' COMMENTS

Reviewer #1 (Remarks to the Author):

The authors have satisfactorily responded to all my questions and made the necessary changes to the manuscript.

Reviewer #3 (Remarks to the Author):

The authors have addressed my comments.

Regarding the Mossbauer part, they gave a satisfactory reply to my comments 1 & 3 by performing a deeper analysis of their data from which they deduce a fluctuation frequency which perfectly matches that deduced from their muon data.

Regarding the muon-site analysis, they took into account my comment 2-a about the H-mu bonding and considered the alternative scenario that I had suggested. This is now very detailed in the supplementary section, the yield is that the two scenarios are possible but do not impact the main stream of the authors' conclusions, obtained using for both a combination of a static fraction and a dynamical one, see below my remaining comment about that fitting procedure.

Yet, they did not answer what I considered as an important issue, comment 2-b, I apologize if I was not clear in my first review. I agree that from the data above T_N , they are led to two fractions, then two muon sites. Can the authors discard a mixed fitting, where

$$A(t)/A_0 = (1-f)*G_{\{3s\}}(t) *G_{\{1,dyn\}}(t)+ f*G_{\{2,dyn\}}(t) \text{ for } T>T_N$$

and

$$A(t)/A_0 = (1-f)* G_{\{3s\}}(t) *G_{\{stat,1\}}(t)*G_{\{1,dyn\}}(t) + f*G_{\{2,stat\}}(t) * G_{\{2,dyn\}}(t).$$

Note that there seems to be some confusion between G_{as} and G_{3s} in the notations used by the authors.

In other words, is it proven that one has to assign a pure static fraction for the muon located at site (1) and set $G_{\{1,dyn\}}$ value to 1 below T_N ? The dichotomy introduced by the authors, ie one muon coupled to static moments and the other one coupled to fluctuating moments, does not seem obvious to me.

I agree that the fitting function that they selected is in-line with the picture suggested from neutron experiments, yet they should rather mention in the supplementary section that they perform a consistent analysis rather than seemingly bringing an additional proof if the above function might fit the data as well. The sentence provided in the main body of the paper (caption of fig.3) goes along that direction "The curve on each dataset represents a fit to a model assuming the coexistence of static and fluctuating local fields"!

We provide point-by-point responses to the referees' comments below. Referees' and our comments are shown in black and blue, respectively.

Reviewer #1 (Remarks to the Author):

The authors have satisfactorily responded to all my questions and made the necessary changes to the manuscript.

We deeply appreciate the reviewer 1's effort to improve our paper.

Reviewer #3 (Remarks to the Author):

The authors have addressed my comments.

Regarding the Mossbauer part, they gave a satisfactory reply to my comments 1 & 3 by performing a deeper analysis of their data from which they deduce a fluctuation frequency which perfectly matches that deduced from their muon data.

We thank to the reviewer for leading us to deeper understanding of Mössbauer spectra and the unified interpretation of the low energy fluctuation.

Regarding the muon-site analysis, they took into account my comment 2-a about the H-mu bonding and considered the alternative scenario that I had suggested. This is now very detailed in the supplementary section, the yield is that the two scenarios are possible but do not impact the main stream of the authors' conclusions, obtained using for both a combination of a static fraction and a dynamical one, see below my remaining comment about that fitting procedure.

We thank to the reviewer 1 for appreciating our analysis of the structure of hydrogen and muon complex.

Yet, they did not answer what I considered as an important issue, comment 2-b, I apologize if I was not clear in my first review. I agree that from the data above T_N , they are led to two fractions, then two muon sites. Can the authors discard a mixed fitting, where

$$A(t) / A_0 = (1 - f) \cdot G_{3s}(t) \cdot G_{1,dyn}(t) + f \cdot G_{2,dyn}(t) \text{ for } T > T_N$$

and

$$A(t) / A_0 = (1 - f) \cdot G_{3s}(t) \cdot G_{1,stat}(t) \cdot G_{1,dyn}(t) + f \cdot G_{2,stat}(t) \cdot G_{2,dyn}(t).$$

We thank to reviewer 3 for clarifying his/her concern. The fitting model the reviewer proposed may be applicable to a situation that both muon sites feel static and fluctuating local fields. We used a simple model in the previous manuscript which assumes that the muon site near a hydrogen feels only a static one and the other muon site feels both static and fluctuating ones. We think that the simple model is better than the proposed mixed fitting as mentioned below.

Regarding the spectra at $T > T_N$, we fitted the spectra using the mixed fitting assuming that the fluctuation rates δ_μ for both sites take a common value. This is because the powder sample consists of pure pharmacosiderite and there is a single magnetic site in the paramagnetic phase. The fitting resulted in a negligible fluctuation rate ν and hence $G_{1,\text{dyn}}(t) \sim 1$, which is exactly the model that we employed in the previous manuscript. As for the spectra below T_N , although we tried to analyse the spectra using the mixed fitting, we were not able to obtain a satisfactory result because of poor convergence. Note that we fixed the f value to that obtained by the fitting above T_N because the population of the two muon sites should not change owing to magnetic ordering. We confirmed that the hyperfine coupling constant and internal magnetic field of the muon site near hydrogen $\mu(1)$ tend to approach small values such that a longitudinal field of 50 mT is sufficient to suppress the spin fluctuation. The results suggest that $\mu(1)$ is insensitive to the spin fluctuation and only the other site $\mu(2)$ can detect it. Thus, our simple model is justified to get reliable information.

In order to make our fitting scheme clear, we added some descriptions in the Supplementary Note 3 as follows.

(Before) Note that the fitting analysis by using $G_{2S}(t)$ instead of $G_{3S}(t)$ gives a slightly larger value of χ^2/N , as show in Figs. S3a and S3b. Thus, we will use the model given by Eq. (2) in the following discussion. The result suggests that there are two muon sites statistically occupied upon muon implantation, which are referred to $\mu(1)$ and $\mu(2)$ for the $(1 - f)$ and f components, respectively.

(After) The fitting analysis using $G_{2S}(t)$ instead of $G_{3S}(t)$ gives a slightly larger value of χ^2/N , as shown in Supplementary Figs. 3a and 3b. In addition, the use of $G_{3S}(t)G_z^{\text{dyn}}(t, \omega_\mu)$ instead of $G_{3S}(t)$ does not improve the fitting. Thus, we use the model given by Supplementary Eqs. (8) and (9) in the following discussion. It is plausible that the observed two magnetic sectors, $(1 - f)$ and f , come from different muon sites, which are referred to $\mu(1)$ and $\mu(2)$, respectively. We note that the parameter f is common in both analyses below and above T_N .

Note that there seems to be some confusion between G_as and G_3s in the notations used by the authors.

This 'G_as' may be $G_{2S}(t)$. We checked that there is no confusion between $G_{2S}(t)$ and $G_{3S}(t)$ in the revised manuscript.

In other words, is it proven that one has to assign a pure static fraction for the muon located at site (1) and set $G_{\{1,\text{dyn}\}}$ value to 1 below T_N ? The dichotomy introduced by the authors, ie

one muon coupled to static moments and the other one coupled to fluctuating moments, does not seem obvious to me.

Yes, it is possible to assign a pure static fraction for site 1 as already answered in the previous section.

I agree that the fitting function that they selected is in-line with the picture suggested from neutron experiments, yet they should rather mention in the supplementary section that they perform a consistent analysis rather than seemingly bringing an additional proof if the above function might fit the data as well. The sentence provided in the main body of the paper (caption of fig.3) goes along that direction “The curve on each dataset represents a fit to a model assuming the coexistence of static and fluctuating local fields”!

We were able to confirm the validity of our dichotomy thanks to the mixed fitting suggested by the reviewer. Without the help of neutron experiments, we can unambiguously interpret the muon data by considering two muon sites $\mu(1)$ and $\mu(2)$ that experience a static field and a dynamic field, respectively. We modified the caption of Fig. 3c as shown below to make our statement clear throughout the paper.

(Before)

The curve on each dataset represents a fit to a model assuming the coexistence of static and fluctuating local fields (Supplementary Note 3).

(After)

The curve on each dataset represents a fit to a model in which one muon site near a hydrogen feels a static local field and the other muon site feels both static and fluctuating local fields (Supplementary Note 3).